# Clinical and Sociodemographic Correlations with Neurological Soft Signs in Hospitalized Patients with Schizophrenia: A Preliminary Longitudinal Study

**DOI:** 10.3390/biomedicines12040787

**Published:** 2024-04-03

**Authors:** Cristian Petrescu, Oana A. Mihalache, Crisanda Vilciu, Diana M. Petrescu, Gabriela Marian, Constantin A. Ciobanu, Adela M. Ciobanu

**Affiliations:** 1Neuroscience Department, Discipline of Psychiatry, Faculty of Medicine, “Carol Davila” University of Medicine and Pharmacy, 020021 Bucharest, Romania; cristian.petrescu@drd.umfcd.ro; 2Department of Psychiatry, ‘Prof. Dr. Alexandru Obregia’ Clinical Hospital of Psychiatry, 041914 Bucharest, Romania; 3Department of Doctoral Studies, “Carol Davila” University of Medicine and Pharmacy, 020021 Bucharest, Romania; oana.mihalache@drd.umfcd.ro; 4Department of Neurology, “Carol Davila” University of Medicine and Pharmacy, 020021 Bucharest, Romania; crisanda.vilciu@umfcd.ro (C.V.); diana-mihaela.vlad@rez.umcd.ro (D.M.P.); 5Neurology Clinic, ‘Fundeni’ Clinical Institute, 022328 Bucharest, Romania; 6Academy of Romanian Scientists, 927180 Bucharest, Romania; gabi.marian@yahoo.com; 7Department of Psychiatry and Psychology, ‘Titu Maiorescu’ University of Medicine, 040051 Bucharest, Romania; 8Faculty of Medicine, “Carol Davila” University of Medicine and Pharmacy, 020022 Bucharest, Romania

**Keywords:** schizophrenia, neurological abnormalities, neurological soft signs, longitudinal assessment

## Abstract

Schizophrenia is a severe, chronic neuropsychiatric disorder characterized by symptoms that profoundly impact behavior, cognition, perception, and emotions, leading to a reduced quality of life and physical impairment. Given the complexity of schizophrenia, there is a pressing need for clinical markers and tools to predict its course, enhance disease staging, facilitate early intervention, improve differential diagnosis, and tailor individualized treatment approaches. Previous studies focused on the relationship between neurological soft signs (NSS) and factors such as age, illness duration, and symptomatology, indicating NSS as state markers improving in parallel with psychotic symptom remission or predicting treatment resistance. However, there is a lack of consensus on NSS assessment tools, hindering routine clinical monitoring despite diagnostic and prognostic potential. The present longitudinal study involved 81 psychiatric inpatients diagnosed with schizophrenia. Patients were assessed at three time points: baseline, 1 month, and 6 months. The examination included the use of scales to evaluate psychotic and neurological symptoms, as well as the identification of adverse extrapyramidal reactions caused by neuroleptic treatment. The progression of NSS was correlated to both the symptomatology and the sociodemographic data of the patients. The main findings from the present investigation revealed a statistical correlation between NSS and psychopathological symptoms, especially with negative symptoms of schizophrenia. However, it is important to note that neuroleptic side effects only had a limited impact on NSS. Therefore, instead of being linked to extrapyramidal symptoms caused by neuroleptics, NSS appears to be more frequently related with symptoms of schizophrenia. Our findings provide further support for their strong association with the course of schizophrenia, independent of treatment side effects, thus emphasizing their potential as reliable assessment tools in both research and clinical settings.

## 1. Introduction

Schizophrenia, a severe chronic neuropsychiatric condition, is distinguished by cognitive, positive, and negative symptoms, and has a profound effect on behavior, perception, and emotions, ultimately resulting in a diminished quality of life and physical impairment [1]. Individuals diagnosed with schizophrenia experience a wide range of outcomes, including a severe chronic course or a partial remission [2]. Social impairments, language difficulties, and motor dysfunctions are prevalent among individuals diagnosed with schizophrenia [3], particularly those who develop the disorder at an early stage [4]. Thus, the importance of clinical evaluation protocol and tools that can be used to predict the course of illness in individuals with schizophrenia or those at a high risk of psychosis arises in order to enhance disease staging, predict and assess symptoms, facilitate early intervention, create a better differential diagnosis, and develop individualized treatment approaches [5].

Schizophrenia is an intricate and diverse condition that consistently generates controversy and debate within the research field. Although the precise cause of schizophrenia is still unknown, current studies provide support for the neurodevelopmental theory [6,7]. The altered development of the brain in schizophrenia may be the result of aberrant gene expression in response to prenatal and perinatal insults, according to this hypothesis [8,9]. The current literature points to the role of motor dysfunction, particularly neurological soft signs (NSS), as a core characteristic of schizophrenia and other related psychotic spectrum diseases [10]. Despite the fact that neurological soft signs (NSS) are not specific to schizophrenia, as these abnormalities are found in healthy population [11], being influenced by age [12], numerous studies have reached the consensus that NSS scores are increased with a marked statistical significance in patients with schizophrenia when compared to either healthy controls [13,14] or their healthy relatives [15,16], in either longitudinal [17] or cross-sectional evaluations [18]. NSS encompasses subtle neurological abnormalities involving sensory integration, motor coordination, and complex motor functions [19]. Regarding the general aspects of NSS in schizophrenia patients, most authors agree with the point that these abnormalities are intrinsic to the disease rather than being side effects of medication [20,21,22], while on the other hand, other researchers reported a correlation between NSS expression and treatment response [23], thus suggesting the assessment of NSS to further predict the response of neuroleptic treatment. Mittal et al. [24] concluded that higher NSS scores predict a poor response to typical antipsychotic treatment, especially haloperidol. Evidence suggests that NSS are more prevalent in schizophrenia patients, neuroleptic-native first-episode patients [25], and high-risk subjects, such as relatives of patients with schizophrenia [26]. Furthermore, NSS correlates with negative symptoms, cognitive impairment, and the risk of psychosis [11], thus NSS may serve as markers of neurodevelopmental abnormalities and might predict the course of a potential psychotic disorder [27,28].

In several previous studies, the main focus was on the relationship between neurological soft signs (NSS) and various factors in schizophrenia, including age, duration of illness, and clinical symptoms, especially negative symptoms of schizophrenia [29,30]. Earlier studies [31] suggested an association between chronic and severe forms of schizophrenia and NSS, while recent studies have largely supported these findings, indicating that NSS may act as state markers that improve with remission of psychotic symptoms [21,32,33] or the use of NSS to predict the emergence of treatment resistance in schizophrenia patients [21]. Patients with a severe course of schizophrenia exhibit a higher prevalence of deficits in motor coordination, sensory integration, and sequencing of complex motor acts [34,35]. As a result of the lack of consensus regarding which of the available tools should be utilized (as the characteristics of these instruments significantly impact study results and conclusions) [36], NSS are not routinely monitored in clinical practice, despite their potential diagnostic and prognostic value [14].

The heterogeneity of NSS assessment items, with regard to biological underpinnings and complexity, contradicts the common belief that NSS rating scales can be used interchangeably to measure the same phenomenon [37]. Variations in the items assessed by these scales may be suggested by discrepancies among clinical signs assessed and subscales. Therefore, it is evident that, to ensure consistent and replicable results in NSS research, it is critical to emphasize the significance of evaluating the degree of concurrence between rating scales [38]. Although various discrepancies in NSS scales have been recognized, a thorough assessment of the manner and degree to which NSS instruments vary in their substance is still lacking [37]. The current literature underscores the need for prospective longitudinal studies to better understand NSS stability across illness stages. In a meta-analysis by Bachmann et al. [39], the authors concluded that during the course of a clinical episode of schizophrenia, the NSS scores had a tendency to decline, particularly as psychopathological symptoms started to improve. When compared to individuals who have non-remitting symptoms of schizophrenia, those who have a remitting course of schizophrenia are more likely to have a decline in their NSS scores. 

The conclusion was based on the fact that the majority of studies included in the meta-analysis reported a decline in NSS scores during the clinical course of illness. However, there were exceptions noted, such as the study by Boks et al. [40], which observed an increase in NSS scores among a group of 29 first-episode patients investigated over a 2-year period. A 2018 review [19] further strengthen previously mentioned hypothesis that irrespective of the number of patients included and the approaches utilized for the evaluation of neuropsychiatric symptoms (NSS), every single study conducted on chronic schizophrenia showed that NSS either maintained a steady course or worsened. On the other hand, investigations conducted on individuals who had a remitting course revealed that NSS reduced throughout the course of time. In their 2022 systematic review of L.E., Pieters et al. [41] discuss the predictive value of neurological soft signs (NSS) in individuals at clinical high risk (CHR) for psychosis and those with first or multiple psychotic episodes. By including 68 articles, the authors also highlight higher NSS levels in schizophrenia patients compared to healthy controls and other psychiatric disorders, also correlating NSS with symptom severity and cognitive dysfunction. To establish the prognostic value of NSS, the authors emphasize the importance of understanding both trait- and state-like features of NSS by pointing NSS scores tendency to decrease with the remission of psychopathological symptoms but remaining higher than healthy controls even after remission, suggesting trait-related characteristics. 

Our study aimed to investigate the relationship between the evaluation of the sensory-motor field (NSS) and psychopathological symptoms in hospitalized patients with schizophrenia over a follow-up period of 6 months. We hypothesized that the NSS scores would change in proportion to the severity of the symptoms and that these scores would not be affected by any treatment. The study also aimed to identify the factors that impact the evolution of NSS scores in relation to scores for neuroleptic-induced parkinsonism and to determine whether certain clinical and demographic factors have an impact on neurological soft signs.

## 2. Materials and Methods

### 2.1. Setting and Subjects

The present longitudinal study comprised 81 psychiatric inpatients (36 males and 45 females) who were admitted to the Prof. Dr. Alexandru Obregia Psychiatry Hospital in Bucharest and consecutively enrolled in the study. The patients’ ages varied from 18 to 64 years (mean age = 33.08, median age = 29.1), and they all fulfilled the diagnostic criteria for schizophrenia as outlined in the DSM V [42]. At the time of enrollment, the patients were prescribed antipsychotic medication by their psychiatrists, without any intervention from the study group in the treatment choice, neither prior to nor subsequent to the assessment. The average daily dose of antipsychotics was 412.04 mg (SD = 196.30) of chlorpromazine equivalent (CPZE) [43,44,45]. Additionally, 22 patients received anticholinergic treatment with trihexyphenidyl at a mean daily dose of 0.67 mg (SD = 1.22) at baseline, which increased to 0.93 mg (SD = 1.46) after 6 months. The antipsychotic dosage remained unaltered for the two weeks preceding the inclusion of the patients in the study. In order to prevent potential interactions with sensorimotor abnormalities, none of the patient received treatment with benzodiazepines. The investigation was granted approval by both the Research Committee and the Ethics Committee of Prof. Dr. Alexandru Obregia Psychiatry Hospital (approval number 89, 7 June 2022). 

A number of the patients included in the study sample were also evaluated in a prior cross-sectional study conducted by our group [46]. A total of 105 patients were initially enrolled in the study. However, during the first assessment, 6 patients were excluded from participation. Furthermore, during the 6-month assessment, an additional 18 patients did not participate, thus the attrition rate was 22.86%, relatively close to other studies with a similar design [5]. The factors leading to the exclusion of the 24 participants were their failure to adhere to the scheduled re-evaluation appointments (14 patients), the requirement for a therapeutic regimen involving high doses of benzodiazepines (8 patients), and voluntary withdrawal from the trial (2 patients). At the second evaluation (one month evaluation), several patients evaluated at baseline were still hospitalized in the same psychiatric hospital. The mean number of hospitalizations during the evaluation was 0.67 (SD = 0.71).

Each participant granted written informed consent following an in-depth description of the study methods, in compliance with the Declaration of Helsinki and the applicable laws of the country. The study’s exclusion criteria included patients who declined participation or failed to provide informed consent, individuals with mental retardation, an organic brain disorder, a history of substance dependence/abuse as defined by DSM V [42], severe head trauma, neurological disorders, or other severe medical conditions that may interfere with the results of the evaluation. Additionally, patients with nonschizophrenia psychotic disorders (such as brief psychotic disorder, schizophreniform disorder, schizoaffective disorder, delusional disorder, schizotypal personality disorder, and affective psychosis), a history of nonpsychiatric drugs with neurological side effects, or those outside the age range of 18–65 were excluded. Patients who missed the follow-up appointment or chose not to continue participating in the study were also excluded.

### 2.2. Measurements

The participants and their relatives were asked to verbally respond to a series of questions from which sociodemographic and medical data were obtained. Additionally, the patients’ medical documents or electronic files were consulted in order to collect further information. The patients’ medical history, years of education, marital status, socioeconomic level, mental and medical history, length of illness, age of disease onset, age of their first hospitalization, the total number of hospitalizations, and past therapies that were delivered were all included in the list of information that was collected.

#### 2.2.1. Assessment of Clinical Symptoms

The Positive and Negative Syndrome Scale (PANSS) was utilized in order to evaluate the clinical symptoms of schizophrenia [47]. Clinical evaluations of patients were carried out on the same day as their neurological evaluations were performed, for every of the three evaluations. The approach established by Leucht et al. [48] was utilized in order to establish a correlation between the PANSS score and the Clinical Global Impression (CGI) Severity score and is presented in Table 1 [49], as well as the CGI improvement score to further correlate the CGI scores with the NSS scores. To avoid underestimating the efficacy of antipsychotic treatment and misinterpreting the correlation between NSS scores and CGI, we subtracted the minimum score of the PANSS (which is considered “no symptoms” at 30) [50] when calculating the percentage reduction.

#### 2.2.2. Neurological Assessment

##### Neurological Evaluation Scale (NES)

At baseline, one month, and six months, the neurological soft signs were evaluated using the Neurological Evaluation Scale (NES) [51]. One subscale of the NES assesses cerebral dominance, tremor, short-term memory, atypical eye movements, and primitive reflexes; the other three subscales measure sensory integration (SI), motor coordination (MC), and sequencing of complex motor acts (SCMA). Its 26 items cover a broad range of neurological manifestations. Except for 2 items, each item is evaluated using a scale from 0 to 2, with 0 being normal, 1 representing a slightly disruptive symptomatology, and 2 representing a very disruptive symptomatology in accordance with its standardized guideline of assessment. To determine the extent of neurological impairment, the total score was calculated along with the values for each of the four subscales.

The Simpson–Angus Extrapyramidal Side-Effects Scale (SAS) [52] was used to evaluate adverse effects from antipsychotic medication such as parkinsonism. This scale has 10 items to rate extrapyramidal side-effects, rated from 0 to 4, with a higher score denoting greater severity. For the present study, a cutoff score of ≥4 on the total SAS score was employed to define parkinsonism [53]. At baseline, 7 patients presented scores above the selected threshold, while at the 1-month and 6-month follow-up, 10 patients registered scores above 4 on the SAS total score, with the highest score of 10 registered in one patient at the 6-month evaluation, representative for a “clinically significant degree of movement disorder” [53].

### 2.3. Statistical Analysis and Data Evaluation

Statistical analysis was conducted using R program version 4.3.2, developed by The R Foundation for Statistical Computing and R Core Team (2023), which provides a language and environment for statistical computing. Additional packages utilized include *lmerTest1* [54], *gtsummary* [55], and *sjPlot*.

## 3. Results

For the present longitudinal, prospective, non-randomized study on a sample of 81 patients with the diagnosis of schizophrenia, the primary endpoint of the study was the NES score, assessed at 3 distinct time points (baseline, after 1 and 6 months), and the secondary endpoint of the study was the SAS score measured at the same 3 distinct time points (baseline, after 1 and 6 months). The socio-demographic characteristics of the study group are presented in Table 2.

The study group consisted of 81 individuals, 45 (56%) females and 36 (44%) males, with a mean age of 33.08 years (SD = 11.32; median = 29.1) and on average, participants had 12.35 years of education (SD = 2.12). Concerning the medical background of the illness, the mean age of psychotic onset was 23.43 years (SD = 5.18; median = 22), while the average duration of illness among participants was 9.65 years (SD = 8.69; median = 6). Participants received their first treatment at an average age of 23.85 years (SD = 5.16; median = 23). Regarding the patient’s need to receive medical assistance in a psychiatric hospital, participants experienced an average of 6.35 hospitalizations (SD = 4.52) from the establishment of the diagnosis with an average total hospitalized period among of 4.83 months (SD = 3.56). During the follow-up, a total of 43 patients were readmitted in the psychiatric department due to either a relapse of psychotic symptoms or for the necessity to modify their course of therapy. The findings from the longitudinal evaluation of the measurement tools employed in the current investigation are displayed along with the Friedman rank *p*-value sum test in Table 3.

The total score and the scores for subscales of the PANSS (Positive and Negative Syndrome Scale)—Positive, Negative, General, and Total—indicate statistically significant changes over time (*p* < 0.001 for PANSS Positive and PANSS General, *p* < 0.001 for PANSS Negative). The antipsychotic treatment expressed in chlorpromazine mg revealed a statistically significant change over time (*p* = 0.023), indicating a variation in medication dosage or type during the study period. The Neurological Evaluation Scale (NES) assessed by subscales of motor coordination, sensory integration, complex motor act sequencing, and the subscale of other signs presented no statistically significantly variation over time, as seen in Figure 1, thus showing that the neurological scores remained generally stable throughout the study period, although less so in the case of NES Motor coordination with a *p*-value = 0.08.

There was no statistically significant change observed in the Simpson–Angus Scale (SAS) throughout the evaluations (*p* = 0.16), indicating that extrapyramidal symptoms induced by neuroleptic treatment were also stable throughout the trial. The design involves correlated measurements (variables were measured at 3 time points in the same patient); therefore, a linear mixed model was used with the dependent variable being the total NES score (or SAS score), and the effects were of two types: fixed for the slope of the predictor coefficients (applicable to all observations) and random (different for each patient) for the intercept in the model for each predictor.

Regarding the administered neuroleptic treatment, the patients received it as follows: (1) initially, 57 patients received neuroleptic treatment using an atypical antipsychotic (Olanzapine, Clozapine, Risperidone, Aripiprazole, Amisulpride, Quetiapine), while 3 patients received haloperidol, and 21 patients received a combination of two atypical antipsychotics. The average daily dose of chlorpromazine equivalent was 412.04 mg (SD = 196.30); (2) at the 1-month evaluation, the number of patients receiving treatment with an atypical antipsychotic increased to 64, with only a single patient receiving haloperidol with the mean daily dose of chlorpromazine equivalent for the study group of 448.46 mg/day (SD = 226.50); (3) following the 6-month examination, 56 patients were undergoing treatment with atypical antipsychotics, 2 with typical antipsychotics (1 patient with haloperidol and another with Zuclopenthixol), and 23 patients received a combination of two antipsychotics. During the final assessment, the average daily dosage of chlorpromazine equivalent was found to be 475.31 mg/day (SD = 243.56), which was the highest among the three assessments. However, it is important to note that this value did not reach statistical significance (*p* = 0.023).

The significance level alpha in the study was 0.05, thus *p*-values less than 0.05 were considered statistically significant. Statistical analysis for the primary endpoint (NES total score) is presented in Table 4. Initially, a simple model with only one predictor was used. Predictors that showed statistically significant influences were then included in a multiple model. The analysis involved a linear regression model that considered both fixed effects (predictors) and random effects (variability among individual patients).

In summary of the statistical analysis based on the NES score, male sex had a statistically significant positive effect (*p* = 0.013) on the NES total score when compared to female patients. Longer illness duration (*p* = 0.029) and more hospitalizations required for the patients (*p* < 0.001) were also statistically linked to higher NES total scores. Regarding clinical assessment scores, CGI scores (*p* = 0.043) and PANSS scores (Positive, Negative, General, and Total) had a statistically significant relationship with the NES total score. Right-hand dominance is associated with a lower NES total score (*p* = 0.011). Higher SAS (Simpson–Angus Scale) scores tend to correlate with higher NES total scores (*p* < 0.001).

Predictors that had statistically significant influences in the simple linear mixed-effects models were introduced into a multiple linear mixed-effects model, followed by the application of a backward selection algorithm used to select the most relevant predictors for inclusion in the model, as presented in Table 5. The goal was to streamline the model by retaining only the predictors that significantly explain the variance in the NES total score.

For the statistical assessment presented in the above table, the R2 for fixed effects was approximately 0.50, R2 for random effects was approximately 0.40, and the conditional R2 (fixed and random effects) was approximately 0.90, with the Random Intercepts SD = 3.1 and Residual SD = 1.5. Thus, the statistical model explains a substantial proportion of the variability in the dependent variable. The residual variability, which represents the portion of the dependent variable not explained by the model, is relatively minimal. 

In our study group, males had a statistically significant effect on the NES total score, with a beta coefficient of 1.6 and a *p*-value of 0.043. Similarly, predictors such as “Duration of illness”, “CGI Improvement”, “PANSS Negative”, “PANSS General”, and the SAS score also demonstrate statistically significant influences on the NES total score.

The analysis for the secondary endpoint (SAS score for neuroleptic induced parkinsonism) involved an initial statistical examination using a simple linear mixed-effects model with a single predictor presented in Table 6. 

A number of associations between the SAS score and certain variables, such as the frequency of hospitalizations during the study, the history of hospitalizations leading up to the initial evaluation, the overall number of hospitalizations, the cumulative time spent in the hospital, and scores on the PANSS Total and subscales (Positive, Negative, General) were noted after the statistical interpretation. In addition, the analysis indicates that there is a slight increase of 0.54 in the SAS score for males compared to females. However, it is important to note that this effect is not statistically significant (*p* = 0.19). There is no significant correlation between age and the SAS score (*p* = 0.96). Similarly, there is no significant association between cerebral dominance (right vs. left) and the SAS score (*p* = 0.32). Increasing the dosage of antipsychotic medication has a notable impact, as indicated by an estimated effect size of 0.17. This means that for every 100 mg increase in chlorpromazine equivalents, the SAS score increases by 0.17. The *p*-value (<0.001) further confirms a statistically significant relationship between EQ CLPZ and SAS score.

According to the findings, individuals who undergo anticholinergic treatment tend to have higher SAS scores compared to those who do not. This association between AC and SAS score is statistically significant, as indicated by the *p*-value (<0.001). Nevertheless, the analysis failed to find a statistically significant association between CGI alone and the SAS score. Statistically significant predictors identified in the simple linear mixed-effects models for the SAS score were incorporated into a multiple linear mixed-effects model. 

Subsequently, a backward selection algorithm was applied in order to systematically remove predictors from the model that do not significantly contribute to explaining the variation in the outcome, thus selecting the most influential predictors for inclusion in the model. The results of the remaining variables with highest statistical correlation to the SAS score are presented in Table 7.

The statistical data presented in Table 7 indicate that the Residual Variance σ2 was determined to be 0.20, and the Random Intercept Variance τ00 had a value of 0.41. This value represents the degree to which the baseline SAS scores differ among individual patients. Higher values indicate a greater variation in SAS scores among patients. The ICC (Intra-class Correlation Coefficient) for the statistical analysis yielded a value of 0.68, signifying the proportion of the overall variance in SAS scores that can be attributed to variations among individual patients. It quantifies the degree of correlation or similarity between SAS scores obtained from multiple measurements within the same patient. To clarify, the ICC value indicates that around 68% of the overall variation in SAS scores can be attributed to variations between individual patients, while the remaining 32% is attributable to variations within patients across multiple measurements. Furthermore, the Marginal R2 and Conditional R2 values were 0.346 and 0.788, respectively. The marginal R-squared value of 0.346 indicates that around 34.6% of the variability in SAS scores can be accounted for by the fixed predictors alone. On the other hand, the conditional R-squared value of 0.788 suggests that when both fixed and random effects (individual differences) are taken into consideration, the model explains approximately 78.8% of the variability in SAS scores.

In summary, the statistical analysis suggests, as expected, that certain patient-specific factors, such as the total number of hospitalizations, treatment response, antipsychotic dosage, and use of anticholinergic medication, are significantly associated with the severity of neuroleptic-induced parkinsonism.

## 4. Discussion

The major findings from the current study include an observed correlation between NSS and poor social functioning and psychopathological symptoms in schizophrenia while neuroleptic side effects only partially contributed to NSS. Thus, rather than being associated with extrapyramidal symptoms induced by neuroleptics, NSS seems to be more commonly associated with symptoms of schizophrenia. Based on the data reported in this current analysis, an increase of 1 point in the SAS score is associated with a mean increase of 0.28 points in the total NES score. Regarding the administered treatment, an increase of 100 mg in chlorpromazine equivalents was associated with an average increase of 0.13 points in SAS while an increase of 1 mg in the dosage of anticholinergics was associated with an average increase of 0.08 points in SAS. Furthermore, an increase of 0.17 points in the SAS was observed for each additional hospitalization. This current view is in line with other previous reports that found no correlation between therapeutical regime independent of the class of antipsychotics and variability of NSS, but rather with the clinical symptoms of schizophrenia, especially with the negative symptoms [12,46,56,57]. This hypothesis is further strengthened by data from various articles that had shown the existence of NSS prior to the initiation of any antipsychotic treatment [15,58].

A notable observation of the present study is the fact that although the scores of the NES subscales were relatively stable during the longitudinal follow-up, without a statistically significant correlation between the three evaluations, the total NES score presented a statistically significant correlation with the PANSS symptomatology evaluation score, especially with the subscale for negative symptoms (an increase of 1 point in the PANSS Negative Score was associated with a mean increase of 0.16 in the total NES score), without any correlation with the dosage of the administered treatment, thus being in line with previous reports [3,46,59].

The current findings of our study generally support the hypothesis of the variation of NSS in parallel with the course of psychopathological symptoms of schizophrenia. The statistical assessment revealed that a mean increase of 2 points in the total NES score is associated with an increase of 1 point in the CGI improvement scale. A significant proportion of studies conducted on individuals diagnosed with schizophrenia demonstrated that neurological soft signs (NSS) have a strong correlation with the intensity and duration of psychopathological symptoms in individuals diagnosed with schizophrenia, showing elevated scores during acute psychotic episodes, that tend to decline with remission [5,12,23,57], while treatment-resistant schizophrenia patients exhibit higher NSS levels compared to treatment-responsive patients [21,34].

In our study group, male gender was associated with a more severe progression, with the average total NES score being 2.7 points higher in male patients compared to female patients. The results of the present study yielded a similar conclusion to other studies regarding the association between NSS expression and male gender [5,12,60]. This association was previously questioned by Prikryl et al. 2007 [61] in a male-only study group of patients with schizophrenia during a 1-year follow-up and further confirmed in a 2012 study [23] carried out by the same authors, which included a 4-year follow-up of male patients with schizophrenia. Even so, there is no unanimous conclusion, as other authors suggested no association of NSS with gender [6,13] while a meta-analysis of Chan et al. [11] could not reach a conclusion due to insufficient data provided by the articles included.

Our data suggest that patients with higher education tend to have a less severe neurological progression, with the average total NES score decreasing by −0.52 for each additional year of education. Individuals with higher education may have more efficient cognitive strategies to cope with the challenges associated with schizophrenia, thus being better prepared to engage in problem-solving, adapt to changes, and utilize compensatory strategies to manage symptoms effectively [62,63]. Also, patients with schizophrenia who are more educated may be more likely to adhere to their prescribed treatment plans, attend therapy sessions, and engage in healthy lifestyle behaviors, which can help mitigate symptom severity and promote overall mental health [64,65]. The present results regarding the correlation of education are in line with those formulated in several other articles [3,12]. Nevertheless, a 2009 meta-analysis [11] concluded that education did not significantly modify the relationships between NSS and the symptom severity of schizophrenia. On the other hand, retired patients experienced a more severe progression compared to other patients, with the average total NES score being nearly 6 points higher than employed patients (statistically significant effect), almost 4 points higher than unemployed patients (statistically significant effect), and nearly 3 points higher than student patients (though the effect is not statistically significant). These patients may experience a significant change in their daily routine and social interactions after retirement due to schizophrenia. Also, reduced social engagement may lead to feelings of isolation, loneliness, financial stress, which can exacerbate symptoms of schizophrenia and contribute to a more severe progression of the illness [66,67]. Retired individuals with or due to schizophrenia may experience a faster decline in cognitive function and overall health compared to their employed or engaged in social activities counterparts, which may lead to a more severe progression of the illness [68]. It is important to mention that, although the present study did not demonstrate significant statistical correlations between the expression of NSS and certain sociodemographic aspects, it is likely that these factors might indirectly influence NSS by affecting the onset and progression of psychotic episodes. Several studies suggested the distribution of psychotic disorders is strongly associated with various social-environmental characteristics such as social isolation of psychotic patients, income inequality, ethnic fragmentation, and physical illness [69,70]. Several authors also examined the relationship between ethnic density and the prevalence of schizophrenia in ethnic minorities [71,72]. It has been shown that when ethnic minorities make up a lower percentage of the local population, the incidence of schizophrenia is higher [73]. Furthermore, researches have provided substantial evidence supporting the strong correlation between exposure to urban environments and the onset of schizophrenia [74]. These studies consistently demonstrate a positive relationship, indicating that the occurrence of schizophrenia tends to increase in a roughly linear ratio as urbanization develops, probably this effect being the result of easier access to mental health services, as other articles demonstrated the correlation between decreased schizophrenia outpatient care in rural areas [75].

Regarding the medical history of the patients, individuals with longer illness duration had, on average, a 0.14 higher total NES score for each additional year, with a 0.37 higher total NES score for each additional hospitalization. Furthermore, an extra month of hospitalization was associated with a 0.52 higher total NES score. Several other authors [14,15,41,46] agree on the fact that patients with longer illness duration often have a more complex and severe symptom profile and a high likelihood of experiencing treatment resistance and recurrent relapses, thus contributing to higher NSS expression.

The present article should be regarded in light of several limitations, such as the relatively small number of patients and selection process bias of including only patients from an emergency psychiatric hospital with most of them having been previously hospitalized, thus meaning that the present study population might present a more severe course of schizophrenia. The small sample size does not allow to generalize the present findings of NSS to the general population, thus extensive investigations including healthy individuals are necessary. The lack of a healthy control group impairs our capacity to make observations on whether the patterns of NSS in schizophrenia patients differ from those observed in healthy people, as our aim was to document changes or patterns of NSS expression over time within a specific schizophrenia patients population. Moreover, the results of the NSS scales should be interpreted with caution due to the potential confounding influence of neuroleptic drug side effects, as those patients usually require a higher dosage or have years of treatment changes that may lead to an increase in side effects. It is important to note that the patient’s exposure to antipsychotic treatment during the 6-month follow-up was heterogeneous, with patients receiving first- or second-class antipsychotics, or in some cases, an association of both. Furthermore, the compliance of patients was only confirmed by following a verbal interview with the patients and caregivers during the follow-up period. Another limitation is the relatively limited number of follow-ups as we faced the difficulty of maintaining regular check-up visits, thus risking a higher drop-out rate. Moreover, the investigators involved in the present analysis were not blinded to the clinical state of the subjects, thus leading to an observer bias in the measurement of NSS.

## 5. Conclusions

While the clinical utility of neurological soft signs (NSS) in patients with schizophrenia remains an area in need of further investigation, our paper contributes to the ongoing debate surrounding the nature and practical value of NSS, as our findings offer additional insights into specific aspects of this debate, further strengthening the strong association of NSS with psychopathological symptoms of schizophrenia and the illness course, without regard to the treatment. Moreover, their clinical relevance has been underscored by numerous longitudinal studies, solidifying their importance in clinical contexts, as NSS can be efficiently and reliably assessed, making them potentially valuable tools in both research and clinical practice, thus creating a great need in a standardized, unanimously accepted NSS assessment with clearly defined cut-off scores. Additional research and validation of NSS, including advanced imaging techniques, are necessary to improve our knowledge and use of these indicators in the staging and management of schizophrenia.

## Figures and Tables

**Figure 1 biomedicines-12-00787-f001:**
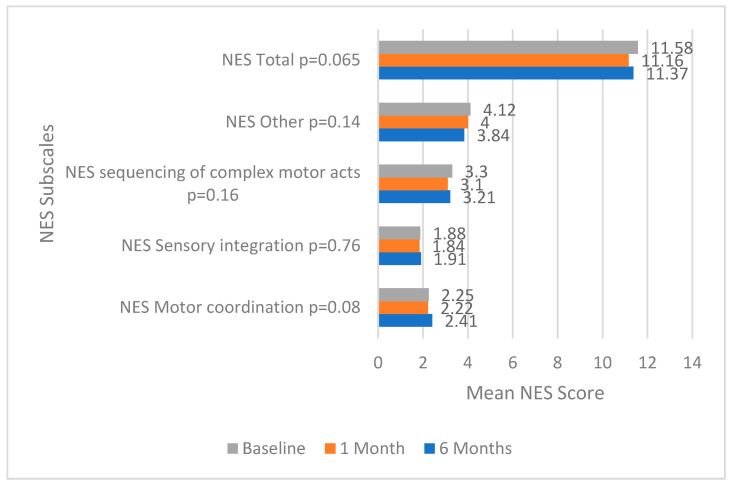
Evolution of NES scores from baseline to 6 months.

**Table 1 biomedicines-12-00787-t001:** Patient’s symptomatology correlated with CGI scores for severity at baseline and improvement after the 6-month follow-up.

**CGI-Severity at Baseline**	**Total N**	**Extremely Ill**	**Severely Ill**	**Markedly Ill**	**Moderately Ill**	**Mildly Ill**	**Borderline, Mentally Ill**	**Normal, Not at All Ill**
Score		7	6	5	4	3	2	1
Patients Group	81	2	13	35	18	11	2	0
**CGI-Improvement after Follow-Up**	**Total N**	**Very Much Worse**	**Much Worse**	**Minimally Worse**	**Unchanged**	**Minimally Better**	**Much Better**	**Very Much Better**
Score		7	6	5	4	3	2	1
Patients Group	81	0	6	10	33	20	12	0

**Table 2 biomedicines-12-00787-t002:** Socio-demographic characteristics of the study group.

Variable	N = 81
Sex, *n* (%)	
Female	45 (56)
Male	36 (44)
Age, Mean (SD)	33.08 (11.32)
Environment, *n* (%)	
Rural	13 (16)
Urban	68 (84)
Marital status, *n* (%)	
Married	18 (22)
Without partner	63 (78)
Years of education, Mean (SD)	12.35 (2.12)
Economic status, *n* (%)	
Retired	37 (46)
Employed	10 (12)
Unemployed	30 (37)
Student	4 (4.9)
Age of onset, Mean (SD)	23.43 (5.18)
Duration of illness, Mean (SD)	9.65 (8.69)
Age at first treatment, Mean (SD)	23.85 (5.16)
Age at first hospitalization, Mean (SD)	24.37 (5.65)
Number of hospitalizations during the follow-up, Mean (SD)	0.67 (0.71)
Number of hospitalizations till first evaluation, Mean (SD)	5.68 (4.40)
Total number of hospitalizations, Mean (SD)	6.35 (4.52)
Cumulative hospitalized period in months, Mean (SD)	4.83 (3.56)
CGI, Mean (SD)	4.66 (1.03)
CGI Improvement, Mean (SD)	3.73 (1.10)
Cerebral Dominance, *n* (%)	
Left	11 (14)
Right	70 (86)

Data are means (SD = standard deviations), unless otherwise indicated.

**Table 3 biomedicines-12-00787-t003:** Summary of scales results and medication from baseline to 6 months.

Variable	Baseline, N = 81	1 Month, N = 81	6 Months, N = 81	*p*-Value *
PANSS Positive				<0.001
Mean (SD)	21.19 (6.01)	19.36 (4.78)	19.28 (5.28)	
Minimum–Maximum	8.00–35.00	8.00–30.00	8.00–31.00	
PANSS Negative				<0.001
Mean (SD)	22.10 (6.78)	20.41 (5.94)	20.89 (6.32)	
Minimum–Maximum	8.00–41.00	8.00–38.00	10.00–38.00	
PANSS General				<0.001
Mean (SD)	42.51 (9.15)	39.69 (8.89)	39.04 (10.26)	
Minimum–Maximum	20.00–65.00	19.00–65.00	19.00–66.00	
PANSS Total				<0.001
Mean (SD)	85.79 (18.02)	79.46 (15.64)	79.21 (18.04)	
Minimum–Maximum	48.00–125.00	50.00–119.00	45.00–121.00	
EQCLPZ				0.023
Mean (SD)	412.04 (196.30)	448.46 (226.50)	475.31 (243.56)	
Minimum–Maximum	75.00–975.00	75.00–1000.00	75.00–1000.00	
NES Motor coordination				0.08
Mean (SD)	2.25 (1.60)	2.22 (1.47)	2.41 (1.44)	
Minimum–Maximum	0.00–8.00	0.00–6.00	0.00–6.00	
NES Sensory integration				0.76
Mean (SD)	1.88 (1.55)	1.84 (1.36)	1.91 (1.43)	
Minimum–Maximum	0.00–7.00	0.00–5.00	0.00–5.00	
NES sequencing of complex motor acts				0.16
Mean (SD)	3.30 (2.03)	3.10 (1.79)	3.21 (1.72)	
Minimum–Maximum	0.00–8.00	0.00–8.00	0.00–7.00	
NES Other				0.14
Mean (SD)	4.12 (2.56)	4.00 (2.26)	3.84 (2.14)	
Minimum–Maximum	0.00–10.00	0.00–9.00	0.00–9.00	
NES Total				0.065
Mean (SD)	11.58 (5.26)	11.16 (4.84)	11.37 (5.13)	
Minimum–Maximum	0.00–22.00	1.00–21.00	0.00–20.00	
SAS				0.16
Mean (SD)	2.90 (2.10)	3.09 (1.90)	3.27 (2.00)	
Minimum–Maximum	0.00–9.00	0.00–8.00	0.00–10.00	

* Repeated Measures ANOVA; SD = standard deviation; EQCLPZ = chlorpromazine equivalent in mg; NES = Neurological Evaluation Scale; SAS = Simpson–Angus Scale; PANSS = Positive and Negative Syndrome Scale.

**Table 4 biomedicines-12-00787-t004:** Statistical analysis for NES total score.

Predictors	N	Beta (95% CI) *	*p*-Value
Sex	81		
Female		—	
Male		2.7 (0.60 to 4.8)	0.013
Age	81 ***	0.09 (0.00 to 0.19)	0.054
Environment	81		
Rural		—	
Urban		−1.1 (−4.0 to 1.9)	0.48
Marital status	81		
Married		—	
Without partner		0.29 (−2.3 to 2.9)	0.83
Years of education	81	−0.52 (−1.0 to −0.01)	0.045
Economic status	81		
Retired		—	
Employed		−5.9 (−9.0 to −2.7)	<0.001
Unemployed		−3.9 (−6.1 to −1.7)	<0.001
Student		−2.8 (−7.5 to 1.9)	0.23
Age of disease onset	81	0.06 (−0.15 to 0.27)	0.57
Duration of illness	81 ***	0.14 (0.01 to 0.26)	0.029
Age at first treatment	81	0.05 (−0.16 to 0.27)	0.63
Age at first hospitalization	81	0.06 (−0.14 to 0.25)	0.56
Number of hospitalizations during the follow-up	81 **	1.3 (−0.19 to 2.9)	0.085
Number of hospitalizations till first evaluation	81	0.37 (0.13 to 0.61)	0.003
Total number of hospitalizations	81	0.38 (0.15 to 0.61)	0.001
Cumulative hospitalized period in months	81	0.52 (0.23 to 0.81)	<0.001
CGI Initial	81	1.3 (0.04 to 2.6)	0.043
CGI Improvement	81 ***	2.6 (1.8 to 3.4)	<0.001
PANSS Positive	81	0.10 (0.03 to 0.17)	0.005
PANSS Negative	81 ***	0.16 (0.09 to 0.24)	<0.001
PANSS General	81 ***	0.10 (0.06 to 0.15)	<0.001
PANSS Total	81 ***	0.06 (0.04 to 0.09)	<0.001
EQ CLPZ	81 ***	0.00 (0.00 to 0.00)	0.12
AC	81 ***	0.08 (−0.11 to 0.27)	0.4
Cerebral Dominance	81		
Left		—	
Right		−4.0 (−7.1 to −0.96)	0.011
SAS	81 ***	0.39 (0.17 to 0.62)	<0.001

* CI = Confidence Interval; PANSS = Positive and Negative Syndrome Scale; EQ CLPZ = chlorpromazine equivalent in mg; AC = anticholinergic treatment; CGI = Clinical Global Impression; SAS = Simpson–Angus Scale; N = number of variables assessed (**—assessed twice; ***—three assessments).

**Table 5 biomedicines-12-00787-t005:** Predictors with statistically significant influences for the NES total score.

Predictors	Beta (95% CI) *	*p*-Value
Sex		
Female	—	
Male	1.6 (0.05 to 3.1)	0.043
Duration of illness	0.13 (0.05 to 0.22)	0.002
CGI Improvement	2.0 (1.3 to 2.7)	<0.001
PANSS Negative	0.11 (0.03 to 0.19)	0.005
PANSS General	0.07 (0.02 to 0.12)	0.005
SAS	0.28 (0.07 to 0.49)	0.01

* CI = Confidence Interval; PANSS = Positive and Negative Syndrome Scale; SAS = Simpson–Angus Scale.

**Table 6 biomedicines-12-00787-t006:** The simple linear mixed-effects model for SAS for extrapyramidal side effects.

Predictors	N	Beta (95% CI)	*p*-Value
Sex	81		
Female		—	
Male		0.54 (−0.27 to 1.4)	0.19
Age	81 ***	0.00 (−0.04 to 0.04)	0.96
Environment	81		
Rural		—	
Urban		−0.32 (−1.4 to 0.79)	0.56
Marital status	81		
Married		—	
Without partner		0.33 (−0.65 to 1.3)	0.51
Years of education	81	−0.20 (−0.39 to 0.00)	0.045
Economic status	81		
Retired		—	
Employed		−1.3 (−2.6 to −0.06)	0.04
Unemployed		−0.88 (−1.8 to −0.01)	0.048
Student		−1.7 (−3.5 to 0.21)	0.081
Age of disease onset	81	−0.05 (−0.12 to 0.03)	0.25
Duration of illness	81 ***	0.02 (−0.03 to 0.06)	0.46
Age at first treatment	81	−0.03 (−0.11 to 0.04)	0.38
Age at first hospitalization	81	−0.02 (−0.09 to 0.06)	0.64
Number of hospitalizations during the follow-up	81 **	0.70 (0.14 to 1.3)	0.015
Number of hospitalizations till first evaluation	81	0.16 (0.07 to 0.25)	<0.001
Total number of hospitalizations	81	0.17 (0.09 to 0.25)	<0.001
Cumulative hospitalized period in months	81	0.21 (0.10 to 0.31)	<0.001
CGI Severity	81	0.34 (−0.06 to 0.74)	0.094
PANSS Positive	81 ***	0.07 (0.03 to 0.11)	0.002
PANSS Negative	81 ***	0.06 (0.02 to 0.10)	0.007
PANSS General	81 ***	0.03 (0.01 to 0.06)	0.01
PANSS Total	81 ***	0.02 (0.01 to 0.04)	<0.001
CGI Improvement	81	0.62 (0.28 to 0.97)	<0.001
EQ CLPZ	81 ***	0.17 (0.13 to 0.20)	<0.001
AC	81 ***	0.33 (0.22 to 0.44)	<0.001
Cerebral Dominance	81		
Left		—	
Right		−0.60 (−1.8 to 0.58)	0.32

CI = Confidence Interval; PANSS = Positive and Negative Syndrome Scale; EQ CLPZ = chlorpromazine equivalent in mg; AC = anticholinergic treatment; CGI = Clinical Global Impression; SAS = Simpson–Angus Scale; N = number of variables assessed (**—assessed twice; ***—three assessments).

**Table 7 biomedicines-12-00787-t007:** Remaining variables with highest statistical correlation to the SAS score.

Predictors	Beta (95% CI) *	*p*-Value
Total number of hospitalizations	0.24 (0.07 to 0.40)	0.005
CGI Improvement	0.18 (0.02 to 0.34)	0.029
EQ CLPZ	0.31 (0.20 to 0.41)	<0.001
AC	0.12 (0.03 to 0.21)	0.009

* CI = Confidence Interval; EQ CLPZ = chlorpromazine equivalent in mg; AC = anticholinergic treatment.

## Data Availability

All the data reported within the article are available in anonymized form upon request from the qualified investigators. The data presented in this study are available on request from the corresponding author.

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
