# Peer review of "Clinical and Sociodemographic Correlations with Neurological Soft Signs in Hospitalized Patients with Schizophrenia: A Preliminary Longitudinal Study"

_biomedicines, 2024, doi:10.3390/biomedicines12040787_

Round 1
Reviewer 1 Report
Comments and Suggestions for Authors
This is a well performed research on neurological soft signs in schizophrenia patients. The sample is somewhat small, but allows some conclusions. However, I could not gather why all the patients had been prescribed with an anticholinergic medication. I gather that could be due to the fact they have been treated with mostly 1st generation antipsychotics, but we do not get a look into the medication break down, just a mean chlorpromazine equivalent. Additionally, we do not know how the patients were chosen for the study (consecutively admitted patients?). At least I could not gather it from the reading, so that should be improved and clarified. Furthermore, why no control group? Let's say early sch vs chronic, or patients without anticholinergic medication vs those with it, etc. Overall, improve the s&m section clarifying these issues, as well as the limitations section. Add preliminary to the title due to the small sample without controls.
Author Response
We are writing to express our sincere gratitude for the time and effort the reviewer dedicated to providing a thorough review of our article. It is our belief that the reviewer’s constructive feedback and insightful comments have been invaluable in shaping the final version of the manuscript. The reviewers’ expertise and attention to detail have undoubtedly enhanced the quality and rigor of the research presented.
In response to the reviewer comments, we have carefully revisited the manuscript and made revisions accordingly. We believe that these changes have strengthened the clarity, coherence, and overall contribution of the article to the field. In the following lines, we provided detailed responses to each of the reviewer points outlining the revisions made and addressing any concerns raised.
The reviewer made reference to the administration of anticholinergics and highlighted the lack of comprehensive presentation about the treatment methods employed for the patients in the text. We find this feedback to be really valuable, hence we would like to notify the reviewer that we have made revisions to the text by including the various forms of therapy delivered : “Regarding the administered neuroleptic treatment, the patients received as fallowing: (1) at baseline, 57 patients received neuroleptic treatment with an atypical antipsychotic (Olanzapine, Clozapine, Risperidone, Aripiprazole, Amisulpride, Quetiapine), other 3 with Haloperidol and 21 with a combination of two atypical antipsychotics with the mean daily dose of Chlorpromazine equivalent of 412.04 mg (SD=196.30); (2) at the 1- month evaluation, the number of patients receiving treatment with an atypical antipsychotic increased to 64, with only a single patient receiving Haloperidol with the mean daily dose of Chlorpromazine equivalent for the study group of 448.46 mg/day (SD=226.50) (3) following the 6-month examination, 56 patients were undergoing treatment with atypical antipsychotics, 2 with typical antipsychotics (1 patient with Haloperidol and another with Zuclopenthixol) and 23 patients received a combination of two antipsychotics. At the final evaluation, the mean daily dose of Chlorpromazine equivalent was 475.31 mg/day (SD=243.56), being the highest of the three evaluations, even so, without a statistical significance (p=0.023). “This modification aims to enhance the transparency of the results. We want to highlight that we did not interfere with the selection of the treatment plan during the assessments.
Regarding the method of patient’s admission into the study, based on the reviewer observation on this missing information, we added that the patients were consecutively enrolled in the study. Please find this modification in the Setting and Subjects subchapter.
The reviewer addressed an extremely important point of this research, namely the lack of a control group for statistical comparison. We consider this point of discussion extremely important, so we want to provide more details that were the basis of this decision. The present study is the continuation of a previous study published in Biomedicines (DOI :10.3390/biomedicines10112939), in which we followed the subgrouping of patients according to the negative symptomatology of schizophrenia, based on the type of treatment, in order to correlate these aspects with the present NSS. Being an observational study, we aimed to investigate changes in NSS within the same group of individuals over time rather than comparing them to a separate control group, so it can provide valuable insights into individual trajectories of development or progression of schizophrenia simptomatology in parralel with NSS. Furthermore, the reviewer made reference to the notion of categorizing patients based on the illness stage or the provided treatment. We would like to bring out to the reviewer the significance of this mention in our future research. However, in order to give the most definitive findings, we require a bigger cohort of patients and an extended duration of follow-up. To summarize this aspect, following the reviewer's advice, we havechanged the title of the article by adding “preliminary” to the title due to the small sample without controls, as well as the limitations section: “The present article should be regarded in light of several limitations, such as the relatively small number of patients and selection process bias of including only patients from an emergency psychiatric hospital with most of them having been previously hospitalized, thus meaning that the present study population might present a more severe course of schizophrenia. The small sample size does not allow to generalize the present findings of NSS to the general population, thus extensive investigations including healthy individuals are necessary. The lack of a healthy control group impairs our capacity to make observations on whether the patterns of NSS in schizophrenia patients differ from those observed in healthy people, as our aim was to document changes or patterns of NSS expression over time within a specific schizophrenia patients population. Moreover, the results of the NSS scales should be interpreted with caution due to the potential confounding influence of neuroleptic drug side effects, as those patients usually require a higher dosage or have years of treatment changes that may lead to an increase in side effects. It is important to note that the patient’s exposure to antipsychotic treatment during the 6-month follow-up was heterogeneous, with patients receiving first- or second-class antipsychotics, or in some cases, an association of both. Furthermore, the compliance of patients was only confirmed by following a verbal interview with the patients and caregivers during the follow-up period. Another limitation is the relatively limited number of follow-ups as we faced the difficulty of maintaining regular check-up visits, thus risking a higher drop-out rate. Moreover, the investigators involved in the present analysis were not blinded to the clinical state of the subjects, thus leading to an observer bias in the measurement of NSS.”
Once again, we would like to thank the reviewer for the conscientious review and valuable feedback, that have been instrumental in improving the manuscript to the current form, and we are confident that the revised version reflects the issues addressed by the reviewer.
We are looking forward to hearing the reviewer’s thoughts on the revised manuscript and we are hopeful for a favorable outcome in the review process.
Reviewer 2 Report
Comments and Suggestions for Authors
The manuscript titled " Clinical and Sociodemographic Correlations with Neurological Soft Signs in Hospitalized Patients with Schizophrenia: A Longitudinal Study" aimed to determine the possible impact of certain clinical and demographic factors on neurological soft signs in hospitalized patients with schizophrenia over a follow-up period of 6 months thereby identifying the factors that impact the evolution of NSS scores in relation to scores for neuroleptic-induced parkinsonism. NSSs are known to be objectively measured, nonlocalizing abnormalities, not related to impairment of a specific brain region, reflecting improper cortical-subcortical and intercortical connections. Previously, a positive correlations betwwen NSS and age and duration of illness in Schizophrenia have been established. The submitted manuscript further elaborates this understanding.
After going through the manuscript, I have following comments for the authors.
1. The age oft he patients ranged from 18 to 64 years with an average age of 33 years. For a better overview oft he age of the patients, please also provide the median age. Similarly, please also mention the median age for disease duration and age of onset.
2. Were the subjects followed up to detect any changes in NSSs over time?
3. As the study is a hospital-based study with small sample size the limitation that the results may not reflect the exact picture of NSS in schizophrenia in the community. Please include this message in the manuscript.
Comments on the Quality of English LanguageEnglish is fine. Minor grammatical corrections and syntax adjustments suggested.
Author Response
We are writing to express our sincere gratitude for the time and effort the reviewer dedicated to providing a thorough review of our article. It is our belief that the reviewer’s constructive feedback and insightful comments have been invaluable in shaping the final version of the manuscript. The reviewers’ expertise and attention to detail have undoubtedly enhanced the quality and rigor of the research presented.
In response to the reviewer comments, we have carefully revisited the manuscript and made revisions accordingly. We believe that these changes have strengthened the clarity, coherence, and overall contribution of the article to the field. In the following lines, we provided detailed responses to each of the reviewer points outlining the revisions made and addressing any concerns raised.
In the first point, the reviewer emphasized the lack of certain statistical data referring to the age and history of the patient's illness. We consider this point of view of the reviewer as a valid one, extremely important, for this reason we want to inform you that we have added all the data requested by the reviewer as follows: “The patients' ages varied from 18 to 64 years (mean age = 33.08, median age= 29.1); mean age of psychotic onset was 23.43 years (SD= 5.18; median=22), average duration of illness among participants was 9.65 years (SD= 8.69; median = 6). Participants received their first treatment at an average age of 23.85 years (SD=5.16; median=23)”. – This data are now presented in the revised form of the manuscript.
Regarding the second issue, the reviewer posted an inquiry regarding patient monitoring in relation to the evolution of NSS. It is important to note that all patients were examined at their initial visit, after one month, and after six months to monitor the progression of NSS using the Neurological Evaluation Scale.
|
|
|
|
|
p-value |
|
NES Motor coordination |
|
|
|
0.08 |
|
Mean (SD) |
2.25 (1.60) |
2.22 (1.47) |
2.41 (1.44) |
|
|
Minimum - Maximum |
0.00 - 8.00 |
0.00 - 6.00 |
0.00 - 6.00 |
|
|
NES Sensory integration |
|
|
|
0.76 |
|
Mean (SD) |
1.88 (1.55) |
1.84 (1.36) |
1.91 (1.43) |
|
|
Minimum - Maximum |
0.00 - 7.00 |
0.00 - 5.00 |
0.00 - 5.00 |
|
|
NES sequencing of complex motor acts |
|
|
|
0.16 |
|
Mean (SD) |
3.30 (2.03) |
3.10 (1.79) |
3.21 (1.72) |
|
|
Minimum - Maximum |
0.00 - 8.00 |
0.00 - 8.00 |
0.00 - 7.00 |
|
|
NES Other |
|
|
|
0.14 |
|
Mean (SD) |
4.12 (2.56) |
4.00 (2.26) |
3.84 (2.14) |
|
|
Minimum - Maximum |
0.00 - 10.00 |
0.00 - 9.00 |
0.00 - 9.00 |
|
|
NES Total |
|
|
|
0.065 |
|
Mean (SD) |
11.58 (5.26) |
11.16 (4.84) |
11.37 (5.13) |
|
|
Minimum - Maximum |
0.00 - 22.00 |
1.00 - 21.00 |
0.00 - 20.00 |
|
Reviewer 3 Report
Comments and Suggestions for Authors
This article presents a topic of interest. Please find below some points for improvement:
The aim of the study seems not to be based on the literature review mentioned above it. Please try to present a more structured literature review.
In addition, there is a discussion not only on symptomatology based on demographics of individuals, but also on ethnic issues on psychoses (see and discuss the points raised in 10.1016/S2215-0366(17)30165-7 ).
The sample size is not justified and is rather small.
Their is a question regarding informed consent. Where the participants with this diagnosis capable to understand and sign their participation?
Please remove the figure from the discussion to the Results section.
The conclusions are not clear and more emphasis should be given to the clinical significance of these findings for the design of assessment protocols for this group of patients.
Comments on the Quality of English LanguageMinor English language editing is necessary.
Author Response
Dear reviewer and editor,
We are writing to express our sincere gratitude for the time and effort the reviewer dedicated to providing a thorough review of our article. It is our belief that the reviewer’s constructive feedback and insightful comments have been invaluable in shaping the final version of the manuscript. The reviewers’ expertise and attention to detail have undoubtedly enhanced the quality and rigor of the research presented.
In response to the reviewer comments, we have carefully revisited the manuscript and made revisions accordingly. We believe that these changes have strengthened the clarity, coherence, and overall contribution of the article to the field. In the following lines, we provided detailed responses to each of the reviewer points outlining the revisions made and addressing any concerns raised.
At first, the reviewer noted an issue in coherence between the Introduction section and the aims of the study. Given the validity of the reviewer’s perspective, we have made modifications to both the introduction and aims in order to present a coherent text that accurately reflects the current understanding of the literature that served as the foundation for formulating the hypotheses of this study. These modifications were made based on the reviewer's suggestion and are now part of the revised version of the manuscript.
Subsequently, the reviewer brought up the matter of ethnic factors in the occurrence and impact on psychoses, citing the article titled "Ethnicity, mortality, and severe mental illness" by Vaitsa Giannouli published in the Lancet Psychiatry journal. We find this approach to be quite fascinating since it offers novel perspectives for future study projects. Simultaneously, in accordance with the reviewer's instructions and utilizing the document provided by the reviewer, I undertook additional documentation on this highly significant topic. This aspect was subsequently addressed in the revised manuscript's discussion section: “It is important to mention that, although the present study did not demonstrate significant statistical correlations between the expression of NSS and certain sociodemographic aspects, it is likely that these factors might indirectly influence NSS by affecting the onset and progression of psychotic episodes. Several studies suggested the distribution of psychotic disorders is strongly associated with various social-environmental characteristics such as social isolation of psychotic patients, income inequality, ethnic fragmentation, and physical illness. Several authors also examined the relationship between ethnic density and the prevalence of schizophrenia in ethnic minorities. It has been shown that when ethnic minorities make up a lower percentage of the local population, the incidence of schizophrenia is higher. Furthermore, researches have provided substantial evidence supporting the strong correlation between exposure to urban environments and the onset of schizophrenia. These studies consistently demonstrate a positive relationship, indicating that the occurrence of schizophrenia tends to increase in a roughly linear ratio as urbanization develops, probably this effect being the result of easier access to mental health services, as other articles demonstrated the correlation between decreased schizophrenia outpatient care in rural areas.”
In the third point of the reviewer, an extremely important point was raised, namely the small number of patients in the cohort, that may influence the final result of the NSS scores. In justifying the present number of patients, we would like to inform the reviewer of the following aspects: 1) for the present study, we used extremely restrictive exclusion/inclusion criteria, perhaps the most restrictive being the exclusion of patients undergoing benzodiazepine treatment in order not to alter the results of the scales for neurological evaluation (NES), but also the exclusion of patients suffering from somatic comorbidities.The study being carried out in a psychiatric emergency hospital, these 2 aspects mentioned above are found very frequently. 2) the patient’s reluctance to participate in research during crisis situations. 3) In an emergency psychiatric hospital setting, patients' conditions can be volatile, and obtaining informed consent and ensuring continuity of participation may pose ethical challenges in a longitudinal study design. 4) It is our view that a small number of participants may be more feasible within the constraints of available resources, ensuring that our team of researchers can adequately monitor and support participants throughout the study duration. 5) Despite the small sample size, our longitudinal study conducted in an emergency psychiatric hospital may provide valuable insights into the trajectory of schizophrenia symptoms in parallel with the presence of neurological soft signs and the treatment outcomes.6) It is our wish that our present study may serve as an exploratory investigation, laying the groundwork for larger-scale research endeavors. Such preliminary findings may shape future research design, sample size estimation, and recruitment strategies. Therefore, based on what was mentioned by the reviewer, we would like to inform you of the following changes: 1) We have modified the Limitation section by adding : “The present article should be regarded in light of several limitations, such as the relatively small number of patients and selection process bias of including only patients from an emergency psychiatric hospital with most of them having been previously hospitalized, thus meaning that the present study population might present a more severe course of schizophrenia. The small sample size does not allow to generalize the present findings of NSS to the general population, thus extensive investigations including healthy individuals are necessary. The lack of a healthy control group impairs our capacity to make observations on whether the patterns of NSS in schizophrenia patients differ from those observed in healthy people, as our aim was to document changes or patterns of NSS expression over time within a specific schizophrenia patients population.” 2) We havechanged the title of the article by adding “preliminary” to the title due to the small sample.
In the next point, the reviewer raised the issue of signing the informed consent by the patients, considering the fact that they suffer from schizophrenia. It is crucial to emphasize that for our research team, obtaining informed consent from individuals with schizophrenia is governed by ethical principles that prioritize respect for autonomy, beneficence, and justice. While schizophrenia may impact decision-making capacity to varying degrees, it is essential to recognize that individuals with mental health conditions still possess the right to participate in research and make informed choices about their involvement, under the current law. Prior to initiating our study, obtaining approval from an institutional review board and ethics committee was essential. The Hospital’s institutional review board,Ethics Committee that included 3 Psychiatry Faculty Professors, one Neurology Faculty Professor and one Clinical Pharmacist and alsoHospital Manager evaluated the ethical soundness of our study protocol, including procedures for obtaining informed consent. We also want to assure the reviewer that the documents mentioned above were sent in their original format, together with a copy translated into English of the informed consent used in the present study, to the MDPI Section Managing Editor at the time of the manuscript submission. Finally, for the drafting of the informed consent, we would like to specify the fact that it was written according to the rigors of the legislation of our country and the legislation of the European Union regarding the rights of patients with mental disorders. Having said that, we would like to thank the reviewer for this vital point of interest in research, namely obtaining informed consent and putting the patient's interests and confidentiality first.
At the reviewer suggestion, we have removed the figure from the discussion to the Results section.
Finally, the reviewer highlighted an issue in the formulation of the conclusions . In light of the reviewer's reasonable remark and our desire to draw accurate conclusions from this paper, we have revised the phrasing of the conclusions in order to provide a more concise summary of the offered information. Please find the following modifications in the new format: “While the clinical utility of neurological soft signs (NSS) in patients with schizophrenia remains an area in need of further investigation, our paper contributes to the ongoing debate surrounding the nature and practical value of NSS, as our findings offer additional insights into specific aspects of this debate, further strengthening the strong association of NSS with psychopathological symptoms of schizophrenia and the illness course, without regard to the treatment. Moreover, their clinical relevance has been underscored by numerous longitudinal studies, solidifying their importance in clinical contexts, as NSS can be efficiently and reliably assessed, making them potentially valuable tools in both research and clinical practice, thus creating a great need in a standardize, unanimously accepted NSS assessment with clearly defined cut-off scores. Additional research and validation of NSS, including advanced imaging techniques are necessary to improve our knowledge and use of these indicators in the staging and management of schizophrenia.”
Once again, we would like to thank the reviewer for the conscientious review and valuable feedback, that have been instrumental in improving the manuscript to the current form, and we are confident that the revised version reflects the issues addressed by the reviewer.
We are looking forward to hearing the reviewer’s thoughts on the revised manuscript and we are hopeful for a favorable outcome in the review process.